# A Robot-Driven 3D Shape Measurement System for Automatic Quality Inspection of Thermal Objects on a Forging Production Line

**DOI:** 10.3390/s18124368

**Published:** 2018-12-10

**Authors:** Liya Han, Xu Cheng, Zhongwei Li, Kai Zhong, Yusheng Shi, Hao Jiang

**Affiliations:** 1State Key Laboratory of Material Processing and Die & Mould Technology, Huazhong University of Science and Technology, Wuhan 430074, China; husthly@foxmail.com or clxyyzxx@hust.edu.cn (L.H.); xuchenghust@foxmail.com (X.C.); shiyusheng@hust.edu.cn (Y.S.); 2Wuhan Vision 3D Technology Ltd., Wuhan 430074, China; jianghao9@126.com

**Keywords:** thermal axles, industrial robot, 3D measurement, 3D data alignment, production line

## Abstract

The three-dimensional (3D) geometric evaluation of large thermal forging parts online is critical to quality control and energy conservation. However, this online 3D measurement task is extremely challenging for commercially available 3D sensors because of the enormous amount of heat radiation and complexity of the online environment. To this end, an automatic and accurate 3D shape measurement system integrated with a fringe projection-based 3D scanner and an industrial robot is presented. To resist thermal radiation, a double filter set and an intelligent temperature control loop are employed in the system. In addition, a time-division-multiplexing trigger is implemented in the system to accelerate pattern projection and capture, and an improved multi-frequency phase-shifting method is proposed to reduce the number of patterns required for 3D reconstruction. Thus, the 3D measurement efficiency is drastically improved and the exposure to the thermal environment is reduced. To perform data alignment in a complex online environment, a view integration method is used in the system to align non-overlapping 3D data from different views based on the repeatability of the robot motion. Meanwhile, a robust 3D registration algorithm is used to align 3D data accurately in the presence of irrelevant background data. These components and algorithms were evaluated by experiments. The system was deployed in a forging factory on a production line and performed a stable online 3D quality inspection for thermal axles.

## 1. Introduction

As robot technology develops, robots replace labor in more and more fields [1]. An ever-growing number of forging tasks are performed automatically by robots. In many cases, the inspection of forgings is still operated by human experts with calipers and inspection templates. However, automated online inspection is needed for the following two reasons:
(1)In an automated forging factory, the first products of a new mold must be inspected to determine if the mold is suitable for mass production. Normally, the inspection is performed after the forgings have cooled. Until the inspection is finished and a report is made, the line must keep the power on standby to avoid out-of-tolerance mass production.(2)During its lifetime, a mold is constantly worn down and the product size gradually changes until tolerances are exceeded. In general, products are sampled and tested after they have cooled. Hence, before out-of-tolerance products caused by excessive mold wear are detected, a large number of products have been manufactured and must be abandoned.

Researchers have developed several kinds of 3D measurement systems to inspect thermal objects over the years. In 2009, Tian et al. [2] integrated time-of-flight equipment and two motors to inspect a large forging cylinder; the system works at a distance of 16 m to remove the influence of high temperature and radiation on the measurement. However, the system generates data point by point and it takes a long time to complete one inspection. In 2011, Du et al. [3,4] improved the aforementioned system by replacing the time-of-flight component with a two-dimensional laser range scanner that generates a line of data at every scan to accelerate the measurement process. Furthermore, in 2012, a shape measurement system with two 3D laser radars was developed by Youssef et al. [5]. It acquires an entire area of point cloud data in one shot, but the data are sparse and only reaches a precision of 6 mm per point. To obtain more accurate data, Liu et al. adopted a high lumen projector [6,7] and line laser array [8] in a system for the triangulation-based measurement of large hot cylindrical forgings. They calculated the diameter of multiple sections instead of obtaining the whole shape data. To acquire more complete 3D data, Zhang et al. [9,10] combined the laser line scanner with a one-dimensional linear motion platform as a full scan system to inspect columnar forgings. However, these applications are only suitable for forgings with simple shapes such as cylindrical shells and columns and are not capable of inspecting relatively complex objects. In recent years, phase measurement profilometry has become increasingly utilized because of its high speed and ability to measure freeform surfaces. In 2015, Zhao et al. [11] used a blue sinusoidal structured light sensor to measure high-temperature objects in 3D, and obtained the dense point cloud data of a small forging part from a fixed view. In their system, optical and digital filters were applied to prevent the influence of visible radiation. In 2018, Li et al. [12] implemented a similar system beside the furnace cavity to monitor selective laser melting additive manufacturing. However, these systems were experimental and did not consider the influence of heat and the need for multi-view measurements. Thus, they are not suitable for online inspection.

In the field of thermal object online inspection, there are two main challenges. The first challenge comes from the high temperature. On one hand, the blackbody radiation disturbs and harms optical equipment, especially when the sensor takes a close-up observation to generate detailed data. On the other hand, the high-temperature surface prevents the use of artificial markers used for data splicing. The second challenge arises from the online environment. First, the inspection time for an online application is limited, especially for thermal parts, whose size varies with time as the temperature decreases. This drives the need for inspection speed and limits the overlapping of data from separate observations. Second, irrelevant background data are inevitable in online environments; for example, it may come from the transferring line or the gripper, which support the parts being inspected.

To this end, an automatic and accurate 3D shape measurement system with heat and radiation resistance is presented. It is integrated with a fringe projection-based 3D scanner and an industrial robot. The system has two advantages:
(1)The measurement speed is increased by the time-division-multiplexing and improved multi-frequency phase-shifting method.(2)Non-overlapping data with irrelevant background data is precisely aligned and registered by the view integration method and the robust ICP-based registration algorithm.

The applied components and algorithms were evaluated by experiments. The system was deployed in a forging factory on a production line and was shown to perform stable online 3D quality inspection for thermal axles.

## 2. Materials and Methods

### 2.1. System Setup

The online inspection system mainly consists of the 3D measurement sensor, cooling system, and industrial robot, as shown in Figure 1. The 3D measurement sensor is composed of two industrial grayscale cameras, one high lumen blue light digital micromirror device projector, and a microcontroller for synchronization. The cooling system includes a closed air circulation loop driven by a pump, a heat exchanger to cool the pumped air to room temperature, and a semiconductor thermoelectric cooling device fixed on the third axis of the robot for further cooling. The 3D measurement sensor is fixed on the end-effector of the industrial robot, connected to the cooling system, and covered by reflective material to reduce incoming heat.

#### 2.1.1. Heat Resistant Design

The radiation wavelengths of thermal objects at temperatures of 1000–1400 K are mainly distributed from 1500 to 2500 nm, leading to a significant thermal effect. Moreover, the visible light radiation reduces the resolution of the projected patterns. A typical blue light filter has high selectivity in the visible light band and can efficiently block the red light of hot objects. However, a visible light filter does not block infrared light, thus allowing infrared light to be focused on imaging elements. To this end, an IR cut filter is placed before the lens of the cameras and projector. As Figure 2 shows, the IR cut filter blocks most of the infrared light while letting through most of the visible light [13].

When the blue light filter and IR cut filter are used, the images of thermal parts become properly exposed and recognizable. Figure 3 shows an image captured by a regular camera and the images captured by the inspection system from one of the views.

Despite the use of the dual filter, the temperature rise of the 3D measurement sensor cannot be ignored. To actively dissipate the heat inside the sensor, the cooling system shown in Figure 4 was designed and implemented. In this system, an air pump circulates the air in a closed loop to avoid dust and condensation. The pump is isolated from the inspection system because of its vibration during operation. The pump compresses the air and raises its temperature, and this is followed by an air-cooling heat exchanger to cool the pumped air to room temperature. Considering the load capacity of the robot axis and the vibration of a compressor refrigeration unit, a solid cooling device with semiconductor thermoelectric components [14] is adopted to further cool the air down by a maximum of 8 °C. The device is mounted on the third axis of the robot to reduce the loss of cold on the way to the sensor.

To keep the temperature inside the sensor stable and maintain the precision of optical devices, thermometers are placed in both the inlet and outlet ports and a temperature feedback pulse-width-modulation control system is adopted [15]. The system keeps the temperature at 30–34 °C in an environmental temperature (away from the thermal parts) of 30 °C during the inspection of thermal parts at 900 °C.

#### 2.1.2. Time Division Multiplexing Trigger

The sensor is based on the synchronization of the projector and cameras. Cameras with network interfaces use jumbo frame data packets for data transmission and prefer exclusive access to the system bandwidth. In a continuous acquisition process, the data captured in one exposure period is transmitted during the next exposure period. This data transmission mechanism works well in a single-camera network. However, when two or more cameras exist in the same network, the jumbo packets interfere with each other and cause frames to be lost. The actual stable bandwidth in such situations is much lower than the physical bandwidth.

To fully use the physical bandwidth of Gigabit Ethernet and reduce the transmission time, a time division multiplexing trigger is employed. As Figure 5 shows, the projector exposure cycle is divided into two equal parts. Each camera is exposed in one part and transmitted in the other part, thus occupying the full bandwidth alternately. While one camera is triggered and exposed, the other camera transmits the data it captured in the last exposure period.

#### 2.1.3. View Integration Method

In normal systems, data can be aligned by attaching artificial markers to the objects or performing an ICP (iterative closest point) algorithm if there is sufficient overlap. However, the high surface temperature of forgings prevents the attachment of artificial markers, and the limited inspection time severely restricts the amount of overlapping area. Thus, alignment based on robot positioning is the only choice.

In traditional systems, a hand-eye calibration is conducted for the robot-sensor system to obtain the coordinate transformation relationship between the scanner and robot end-effector. With the transformation from robot end-effector coordinate system to robot base coordinate system (provided by the robot controller), a coordinate conversion chain is formed from the scanner to the robot base. Data observations from different views can be aligned in the robot base coordinate system. However, the coordinates of end-effector provided by the robot controller are not accurate. In fact, the absolute precision provided by common commercial robots is usually too poor for an accurate hand-eye calibration [16,17]. Instead, the repeatability of the robot’s taught movements is sufficiently accurate because the angle of each joint of the robot is obtained by the encoder and is controlled by a closed loop [18,19]. Hence, we developed the following solution for data fusion.

First, artificial markers are attached to a target that has a shape similar to that of the scanned part. The target is scanned by a portable photogrammetry camera to generate an accurate points’ array of the artificial markers, as shown in Figure 6. Then, the target is placed on the inspection station instead of the part to be scanned. The system is switched to recording mode and runs the same path as it does when scanning a real part. In each limited view, the scanner identifies the artificial markers and generates their 3D coordinates. Then they can be located in the points’ array by their unique topological relationship, and iteratively aligned to the points’ array. When the iterative alignment reaches its desired precision, the final matrix is recorded and associated with this view. When the recording scan is over, each view has a matrix for converting the captured data to the points’ array’s coordinate system.

During a regular scan, the data of each view is converted using the corresponding matrix and aligned to the points’ array’s coordinate system. The precision of the alignment is guaranteed by the repeatability of the industrial robot and satisfies the requirements for vehicle axle inspection. Furthermore, the points’ array’s coordinate system is pre-aligned with the design model, thus providing a pre-calibrated rigid transformation for the registration (see Section 2.2.2).

In order to minimize the effects of robot body temperature change on robot repeatability, the recording process is performed after the robot body has warmed up to the average temperature when production line runs stably.

To validate the repeatability of the robot-sensor system, which is critical in this view integration method, a target consisting of a pair of ceramic balls was repetitively approached and scanned by the system 10 times. Two metrics were adopted: the envelope sphere radius of the centers to evaluate positional repeatability, and the envelope cone angle of center line vectors to evaluate orientational repeatability. In this experiment, they are 0.011 mm and $1.550\times 10^{-4}\;\mathrm{rad}$.

In addition, the absolute accuracy in this experiment is 0.030 mm (at a distance of 150.124 mm) according to the comparison of ball center distances. The true value of the ball center distance comes from a coordinate measuring machine (model: HEXAGON GLOBAL CLASSIC SR 07.10.07, HEXAGON, Stockholm, Sweden).

### 2.2. Algorithms

#### 2.2.1. Improved Multi-Frequency Phase-Shifting Method

The scanning speed of the system is constrained by the flip frequency of the digital micromirror device and CCD exposure frame rate. In this situation, a feasible method for accelerating the process is to reduce the number of projected patterns.

Phase-shifting algorithms are widely used in optical metrology because of their measurement speed and accuracy [20]. Usually, multi-frequency heterodyne technology [21] is used to overcome the π phase discontinuities in phase shifting and obtain the absolute phase map. In this paper, we used three frequencies in a four-step phase-shifting method to balance the cost of pattern acquisition and error resistance [22].

Typically, a serial of sinusoidal fringe images with a constant amount of phase-shifting is projected on the target surface and the two cameras capture the distorted fringe images synchronously [23]. In particular, the images captured by the cameras can be expressed as
(1)$$I_{ji}\left(x,y\right)=A\left(x,y\right)+B\left(x,y\right)\cos\left[\varphi_{j}\left(x,y\right)+\delta_{ji}\right],$$
where (x,y) denotes the pixel coordinates, which can be omitted in the following expressions, $I_{ji}$ is the recorded intensity of the ith image of the jth frequency, A is the average intensity, B is the modulation intensity, $\delta_{ji}$ is the constant phase-shift, and $\varphi_{j}\left(x,y\right)$ is the desired phase information of the jth frequency.

In the multi-frequency phase-shifting method, the information is redundant if ambient light image A can be regarded as a constant [23]. For the N-step phase shifting pattern sequence, knowing that $\delta_{ji}=2\pi i/N$, we can easily obtain
(2)$$\sum_{i=1}^{N}I_{ji}=NA,$$

In particular, when N is even,
(3)$$I_{ji}+I_{j\left(i+N/2\right)}=2A,$$

This means that the ith image and the (i+N/2)th image of the same frequency have complementary phases and can be reproduced by each other.

Hence, the patterns needed for an m-frequency N-step phase-shifting method can be reduced from mN to mN/2+1 when N is even.

Specifically, for the three-frequency four-step phase shifting used in this paper, N=4, i={1,2,3,4} and j={1,2,3},
(4)$$\phi_{j}=-\arctan\frac{\sum_{i=1}^{4}I_{ji}\sin\left(\delta_{ji}\right)}{\sum_{i=1}^{4}I_{ji}\cos\left(\delta_{ji}\right)}=\arctan\frac{I_{11}+I_{13}-2I_{j2}}{2I_{j1}-I_{11}-I_{13}},$$

The number of patterns is reduced from 12+1 (a pattern sequence of 4–4–4–1) to 7+1 (a pattern sequence of 3–2–2–1), where 1 represents a reference pattern with plain illumination. Thus, the process time is reduced by 38.5%.

To evaluate this approach, objects were measured both by the original multi-frequency phase-shifting method and the improved method. In this experiment, we reconstructed the 3D data respectively from the whole set of patterns and a selected set from the whole set which only contains the specific frames mentioned in the algorithm. Figure 7 shows the 3D comparison results of the original method and the improved method. The maximum allowable deviation between the two results in this experiment was 0.02 mm, and the points within this deviation are shown in green. Most of the points from the improved method match those of the original method to an accuracy sufficient for key dimension fitting and extraction.

#### 2.2.2. Noise-Insensitive Data Registration

After transforming multi-view data into the points’ array’s coordinate system, the fused point cloud data needs to be further registered with the design model for structure segmentation and feature extraction. For the instability of the transferring robot’s gripper [24] and the non-positioning bracket on the transferring line, the pose of the workpiece after being grasped and placed is slightly different every time. In this situation, the fused point cloud data is not exactly in the same position and orientation every time. To this end, a coarse-to-fine registration strategy is adopted in which a pre-calibrated rigid transformation is used for a rough alignment followed by the ICP algorithm for further pose optimization. However, because this is an online system (Figure 8) and the inspection station is, in fact, just a specific location on the transferring line, some surface data from the transferring line are inevitably included in the point cloud data during the measurement process. When using the ICP algorithm to perform the fine registration, these invariant/noise data will prevent the correct alignment of the fused data and design model.

Some variants of the ICP algorithm were tested to reduce the sensitivity of the ICP to invariant/noise data. Because the object on the inspection station is not completely constrained and positioned, after the course registration with the pre-calibrated rigid transformation, the position of the data relative to the design model is not fixed. It is hence difficult to set a suitable constant distance threshold to reject incorrect correspondence [25]. Moreover, using Trimmed-ICP to set a constant inlier percentage rate at which to reject incorrect correspondences has been proven to fail [26]. To solve this problem, we propose an adaptive distance threshold-based ICP algorithm. The distance threshold in each iteration is dynamically updated by performing a statistical analysis of the nearest neighbor distances between each matching point.

To accelerate the calculation of point pair matches, a KD-tree is used. Simultaneously, a point-to-surface error metric with a better anti-interference performance for point cloud noise and outliers is adopted in the optimization method. Specifically, given measurement point cloud P and design model point cloud Q, for each point $p_{i}$ in P, we find its nearest neighbor $q_{i}$ in Q, and the solution for the objective function of the corresponding rotation and translation matrices is
(5)$$F\left(R,T\right)=arg\min\sum_{i=1}^{N_{c}}{\left(\left(R_{p_{i}}+T-q_{i}\right)\cdot n_{i}\right)}^{2},$$
where $N_{c}$ is the number of elements of the matching points set$\;C=\left\{\left(p_{i},\;q_{i}\right)\right\}$ and $n_{i}$ is the normal vector of $q_{i}.$

The flowchart of the algorithm is shown in Figure 9, and the specific implementation steps are as follows:

Step 1. Input measurement point cloud P and design model point cloud Q.

Step 2. Create a KD-tree of Q for nearest neighbor search and calculate the resolution r of the model point cloud (used to set convergence conditions).

Step 3. Traverse all points $p_{i}\;$in P and search for the closest point $q_{i}\;$in Q according to the KD-tree; these two points are the matching point pair set $\left\{C_{i}\right\}$.

Step 4. Calculate the Euclidean distance$\;d_{i}$ of all point pairs in$\;\left\{C_{i}\right\}$. With their average distance $\bar{d}$ and standard deviation σ computed, we define the adaptive distance threshold τ as
(6)$$\tau =\bar{d}+a\cdot \sigma ,$$
where a is a scalar factor controlling the threshold range. Each matching point pair whose Euclidean distance $d_{i}$ lies outside the adaptive distance threshold τ is removed from the matching points set $\left\{C_{i}\right\}$. Sets $\left\{C_{i}\right\}$ and P are updated to $\left\{C_{i}^{\prime }\right\}$ and $P^{\prime }$.

Step 5. Sort points pairs in$\;\left\{C_{i}^{\prime }\right\}$ by $d_{i}$ and take the top 70% of the data as a basis for calculation. Sets $\left\{C_{i}^{\prime }\right\}$ and $P^{\prime }$ are updated to $\left\{C_{i}^{\prime \prime }\right\}$ and$\;P^{\prime \prime }$.

Step 6. According to the external rotation order X−Y−Z, set the rotation angle to be α,β,γ and find
(7)$$R=R_{z}\left(\gamma \right)R_{y}\left(\beta \right)R_{x}\left(\alpha \right)=\left[\begin{array}{ccc} r_{11} & r_{12} & r_{13}\\ r_{21} & r_{22} & r_{23}\\ r_{31} & r_{32} & r_{33} \end{array}\right],$$
in which

$r_{11}=\cos\gamma \cos\beta ,r_{12}=-\sin\gamma \cos\alpha +\cos\gamma \sin\beta \sin\alpha ,\;r_{13}=\sin\gamma \sin\alpha +\cos\gamma \sin\beta \cos\alpha ,$$r_{21}=\sin\gamma \cos\beta ,r_{22}=\cos\gamma \cos\alpha +\sin\gamma \sin\beta \sin\alpha ,r_{23}=-\cos\gamma \sin\alpha +\sin\gamma \sin\beta \cos\alpha ,\;r_{31}=-\sin\beta ,$ and $r_{32}=\cos\beta \sin\alpha ,\;r_{33}=\cos\beta \cos\alpha ,$

Suppose that α,β,γ are close to 0. Then, we have sinθ≈θ,cosθ≈1. Let $T={\left(t_{x},t_{y},t_{z}\right)}^{T}$, and in this situation, Equation (5) becomes
(8)$$\begin{array}{c} F\left(R,T\right)=F\left(\alpha ,\beta ,\gamma ,t_{x},t_{y},t_{z}\right)\approx \\ \arg\min{\left({\left[\begin{array}{ccc} p_{1}^{\prime \prime }\times n_{1}^{\prime \prime } & \cdots & p_{N_{C^{\prime \prime }}}{}^{\prime \prime }\times n_{N_{C^{\prime \prime }}}{}^{\prime \prime }\\ n_{1}^{\prime \prime } & \cdots & n_{N_{C^{\prime \prime }}}{}^{\prime \prime } \end{array}\right]}^{T}\left[\begin{array}{c} \begin{array}{c} \alpha \\ \beta \\ \gamma \end{array}\\ \begin{array}{c} t_{x}\\ t_{y} \end{array}\\ t_{z} \end{array}\right]-\left[\begin{array}{c} n_{1}^{\prime \prime }{}^{T}\left(q_{1}^{\prime \prime }-p_{1}^{\prime \prime }\right)\\ \cdots \\ n_{N_{C^{\prime \prime }}}{}^{\prime \prime T}\left(q_{N_{C^{\prime \prime }}}{}^{\prime \prime }-p_{N_{C^{\prime \prime }}}{}^{\prime \prime }\right) \end{array}\right]\right)}^{2}, \end{array}$$

Then, $\alpha ,\beta ,\gamma ,t_{x},t_{y},$ and $t_{z}$ can be solved using the SVD method [27] and R and T are obtained.

Step 7. If the convergence condition is satisfied, the algorithm ends; otherwise, the process proceeds to Step 3 and the next iteration is performed.

The root-mean-square distance convergence condition is adopted in the traditional ICP algorithm. However, because the data object processed by the algorithm is point cloud data with unrelated points, there is a case in which the registration actually converges, but the root-mean-square value remains large because some unrelated points have not been completely eliminated. In this case, it is determined that convergence has not occurred and the iterations are continued. Therefore, in the system proposed in this paper, a different convergence condition is employed.

In the proposed method, rotation and translation convergence conditions are used. When the algorithm satisfies both conditions, the registration is found to have converged. Moreover, the maximum number of iterations of the algorithm is limited to 30.

The rotation convergence condition is defined as whether the cosine of the iteration’s rotation angle θ is greater than a threshold e (cosθ>e). In this method, the value of e is 0.99999, which means the rotation convergence condition is satisfied when θ is less than 0.256°. Hence, according to the angle axis representation method of rotation matrix R
(9)$$\cos\theta =\frac{\left(r_{11}+r_{22}+r_{33}-1\right)}{2},$$

The translational convergence condition is defined as whether the translation distance D of the iteration is less than threshold t (D<t). In this method, the value of t is adaptively obtained according to the point cloud resolution r of Q (Equation (10)). Point cloud resolution is defined as the mean of the distance between the points in the point cloud and its nearest point, reflecting the density of the point cloud. Hence,
(10)$$t=\tau r,r=\frac{1}{N_{q}}\sum_{i=1}^{N_{q}}\|q_{i}-q_{i-closest}\|,$$
where τ is a scale factor that can be adjusted according to the required accuracy. The value of τ in this study is 0.6, $N_{q}$ is the number of points in Q, and $q_{i-closest}$ is the nearest point of $q_{i}$ in Q.

To demonstrate the performance of the proposed method, it is comp ared with Trimmed-ICP. Two workpieces with different backgrounds were each scanned and aligned to their design models. Figure 10 shows the results of the two methods, in which scanned data are colored blue, invariant/noise data are gray, and the design model is green.

## 3. Results

### 3.1. Overview of the Work Site

To evaluate the system in a real environment, the system was placed beside the transferring line from the forging press to the sorting area (Figure 11). When the thermal axle reached the inspection station, a signal was given and the industrial robot traversed all positions scheduled in the calibration procedure. At each position, the sensor took a series of pictures illuminated by the projector and generated the point cloud data.

The axles are produced at a rate of one every 45 s. The scanning procedure costs 27 s for each axle and the 3D data processing costs 12~13 s. Taking into account the time of the transferring line movement, the data processing procedure is executed in parallel. The transferring line has 4 stations and an axle reaches the final station in two cycles after being scanned (Appendix A). The inspection result of each axle is returned before it reaches the end of the line so it can be sorted into different post-processing areas by the robot.

We used the AutoScan^®^ software developed by Wuhan Vision 3D Technology Ltd in Wuhan, China to generate point cloud data from camera frames, align them to the coordinate system of markers points’ array and register the aligned data to the design model with the proposed noise-insensitive data registration method. Then the point cloud data was processed automatically in the Geomagic Control X^®^ software, which extracts the dimensions from the point cloud data and generates a report for each data. Figure 12 shows the data acquisition and processing mechanism of the system, in which the serial execution part and parallel execution part is distinguished by red dotted line.

### 3.2. Experiments

To verify the methods proposed in this paper, two experiments were conducted. First, the overall data alignment was tested several times to estimate its stability and then the whole system was operated within a production line to prove its feasibility.

#### 3.2.1. Evaluation of the Precision of 3D Data Alignment

A cooled vehicle axle was repeatedly inspected on the transferring line several times and the key dimensions were extracted according to the segmentations on the designed model. Invariant/noise data occupied 22% of the original scanned data in this situation. As the results in Figure 13 show, invariant/noise data were removed after the registration and the processed data pieces were correctly aligned to the design model. The results listed in Table 1 further show that the maximum deviation for the total length is 0.11 mm, which is in line with expectations (1.0 mm in total).

#### 3.2.2. Validation of the System

For the overall evaluation, the system worked within the production line and inspected several high-temperature vehicle axles. The system worked well and the key dimensions were correctly obtained. Figure 14 shows three examples of the online inspection data processing in which the yellow areas represent the aligned 3D point cloud data.

Moreover, the trends in the key dimensions can be traced synchronously. As Figure 15 shows, in a pilot production bunch of nine workpieces, the key dimensions clearly decrease as the products are processed.

## 4. Conclusions

In this paper, a 3D shape measurement system for online inspection of thermal objects was introduced. Innovative methods for rapid 3D inspection and data alignment in an online situation were proposed and tested. The use of an improved multi-frequency phase-shifting method reduced the time required for a typical four-step three-frequency phase-shifting operation by 38.5%. The robust data registration algorithm aligned data pieces to the design model accurately when approximately 22% of the data was irrelevant.

Experiments further demonstrated the feasibility of the approach. The system worked well in the online inspection of thermal axles and demonstrated potential for trend analysis. The system design for heat resistance, image transmission, and data alignment could also provide solutions for similar applications.

It should be noted that the proposed view integration method is less flexible and needs to be performed separately for different plans and when environmental temperature changes.

It should also be noted that there is still much room for improvement in this work. In such an application scenario, the means of estimating absolute accuracy is to be discovered. Long term quantitative assessment of robotic repeatability in thermal state also needs to be completed. In future work, we plan to use targets whose thermal dimensions can be accurately obtained as the tool for accuracy evaluation. We are also experimenting with other precision assurance and verification methods for comparison.

## Figures and Tables

**Figure 1 sensors-18-04368-f001:**
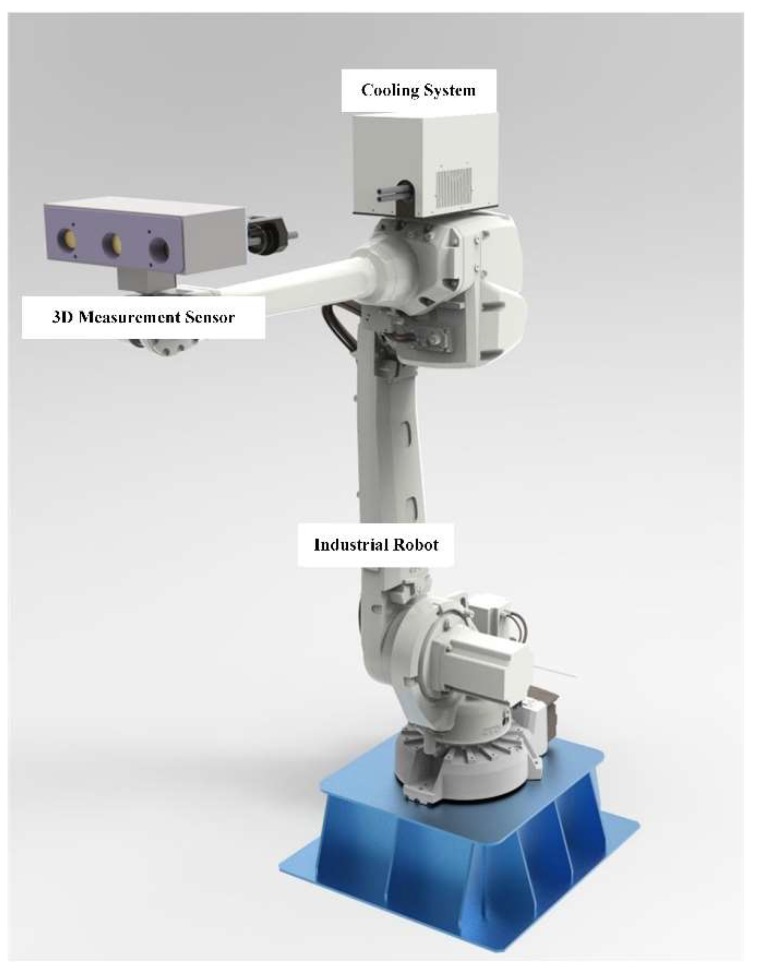
Overview of the online inspection system.

**Figure 2 sensors-18-04368-f002:**
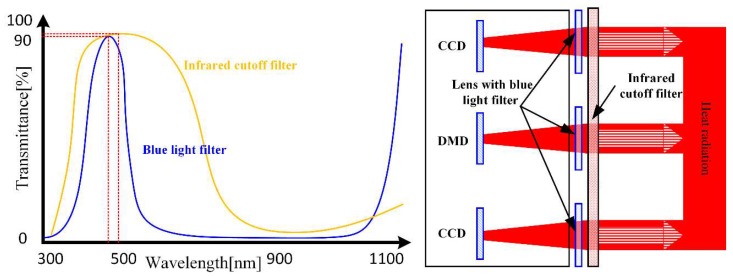
Transmittance spectrum of the blue light filter and IR cut filter (**left**) and their arrangement (**right**).

**Figure 3 sensors-18-04368-f003:**
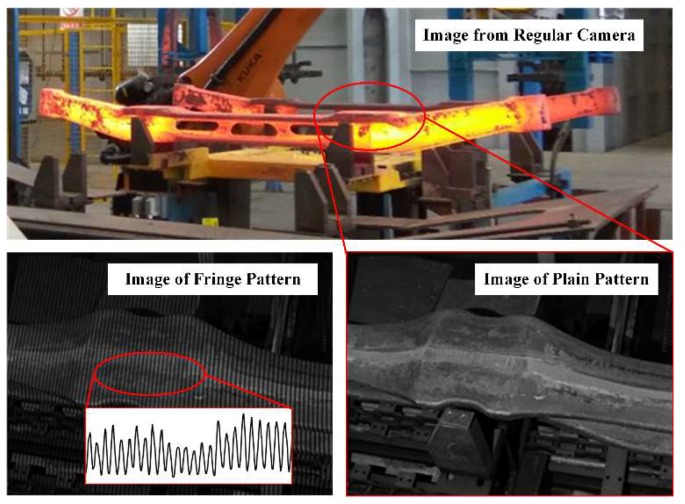
Images captured by a regular camera (**top**) and the inspection system (**bottom**).

**Figure 4 sensors-18-04368-f004:**
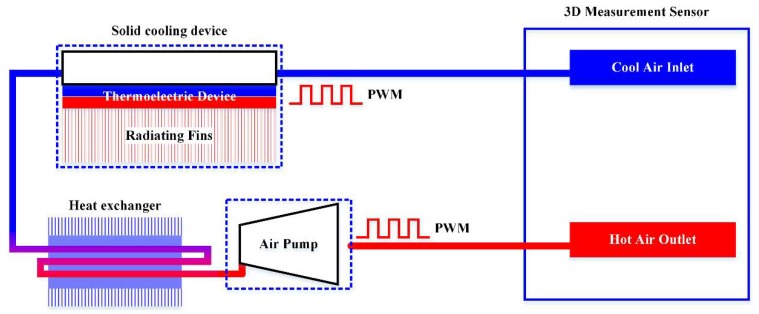
Air circulation loop.

**Figure 5 sensors-18-04368-f005:**
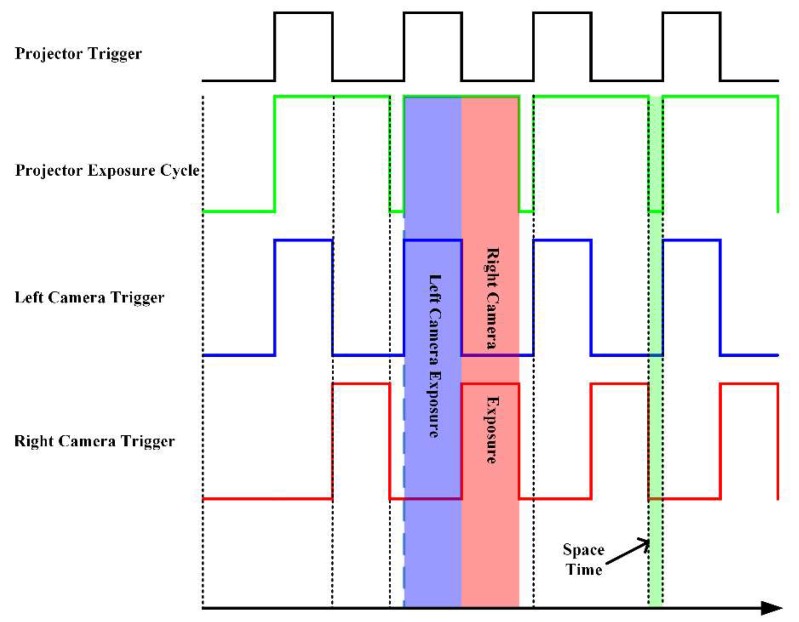
Time division multiplexing trigger timing diagram.

**Figure 6 sensors-18-04368-f006:**
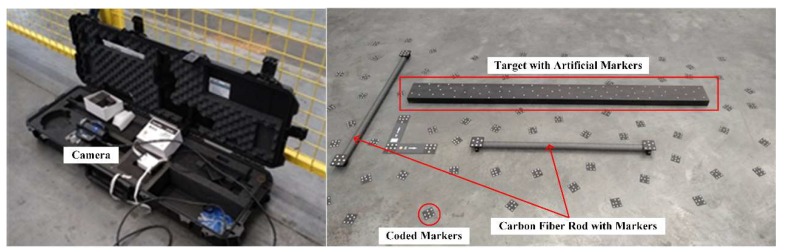
Equipment for calibrating the target and the calibration setup.

**Figure 7 sensors-18-04368-f007:**
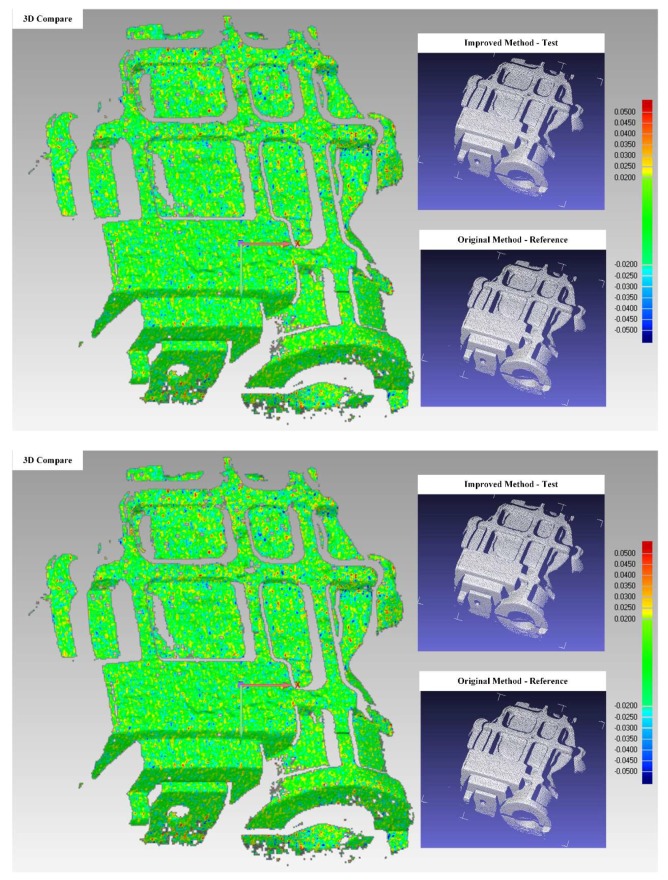
Improved multi-frequency phase-shifting method comparison.

**Figure 8 sensors-18-04368-f008:**
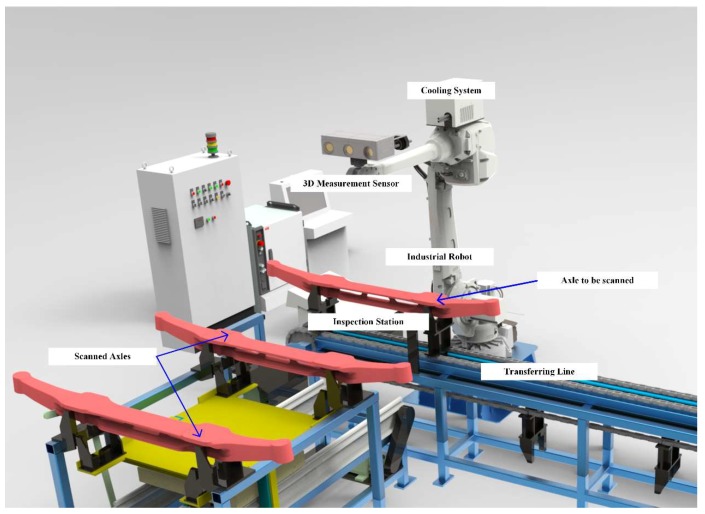
Three-dimensional schematic diagram of the work site.

**Figure 9 sensors-18-04368-f009:**
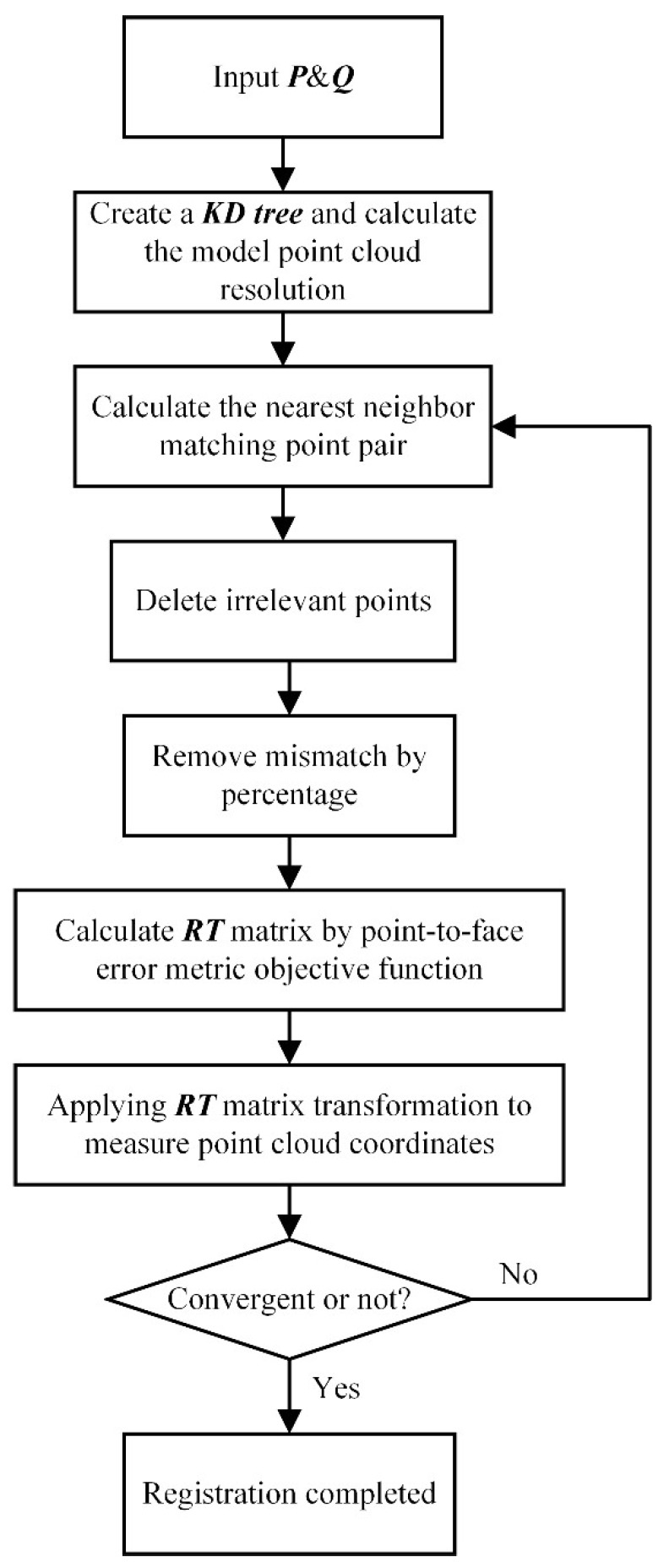
Flowchart of the adaptive distance threshold-based ICP algorithm.

**Figure 10 sensors-18-04368-f010:**
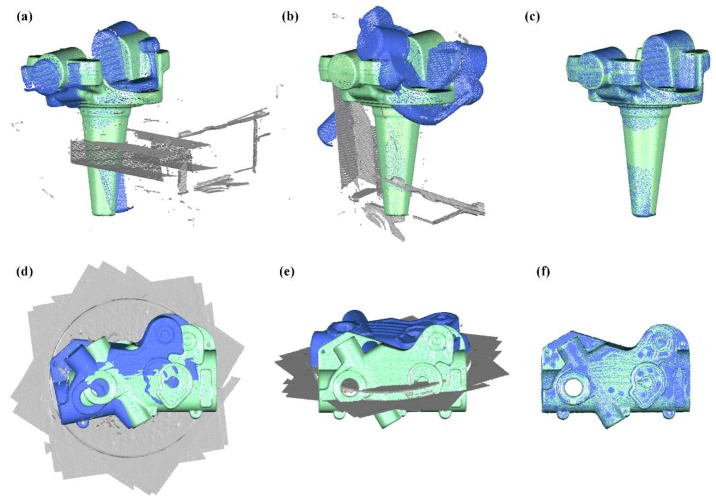
Data alignment experiment. (**a**) original data positions of a knuckle with the background data of a clamping device; (**b**) alignment result of Trimmed-ICP; (**c**) alignment result of our method; (**d**) original data positions of a cover with background data of a platform; (**e**) alignment result of Trimmed-ICP; and (**f**) alignment result of our method.

**Figure 11 sensors-18-04368-f011:**
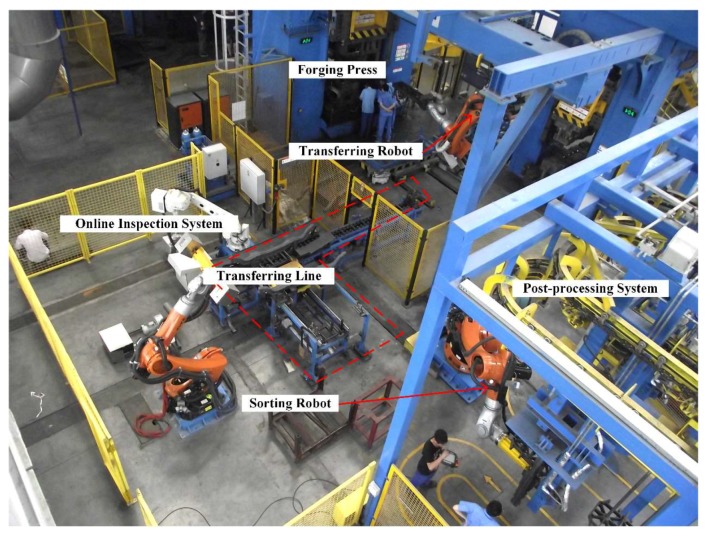
Overview of the work site.

**Figure 12 sensors-18-04368-f012:**
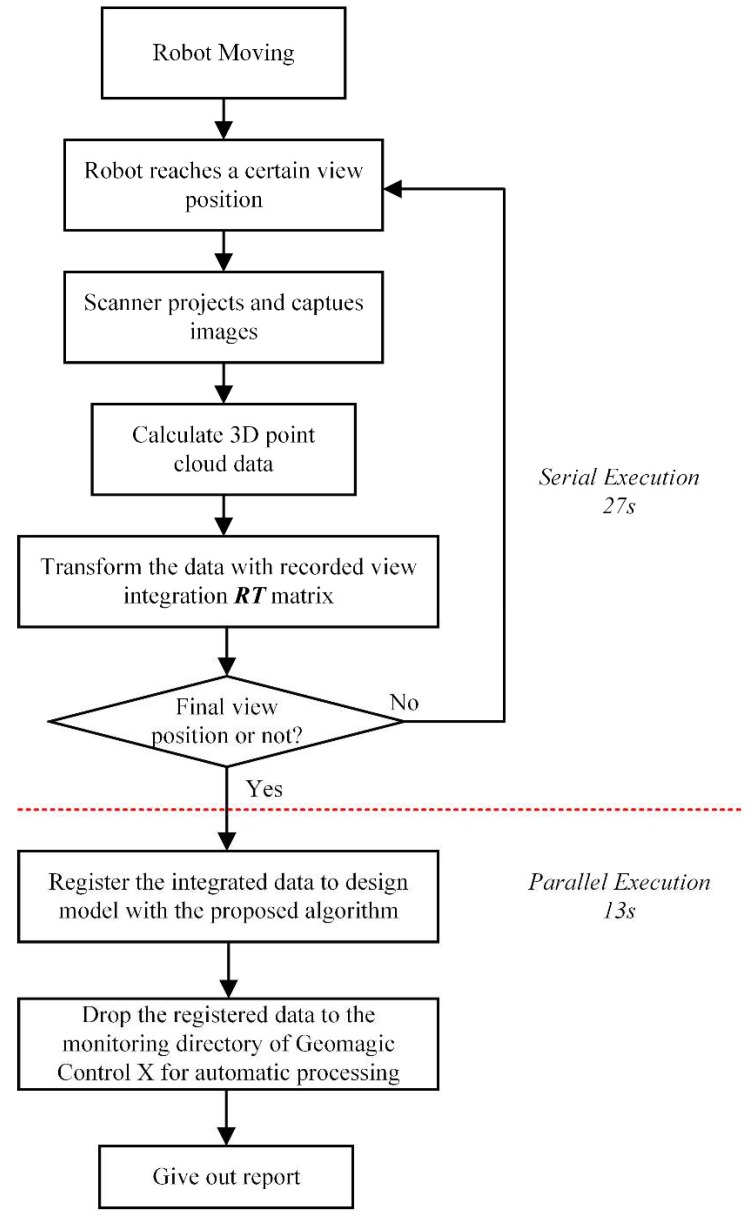
Data acquisition and processing mechanism.

**Figure 13 sensors-18-04368-f013:**
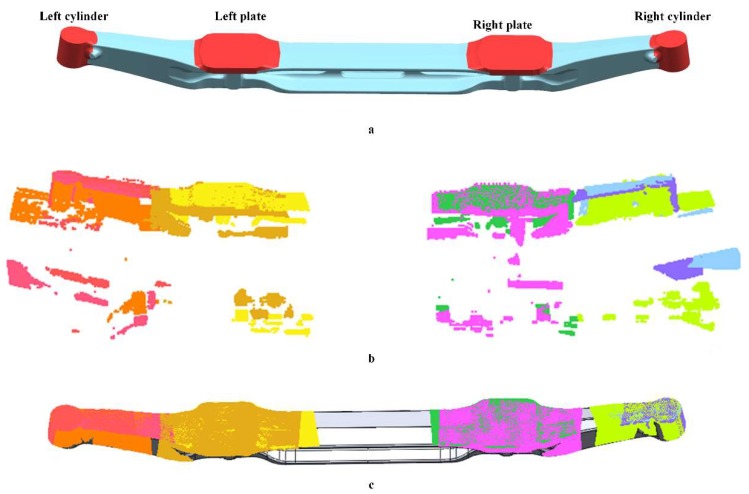
Data alignment of a vehicle axle. (**a**) Segmentation of the designed model; (**b**) original scanned data; and (**c**) aligned data.

**Figure 14 sensors-18-04368-f014:**
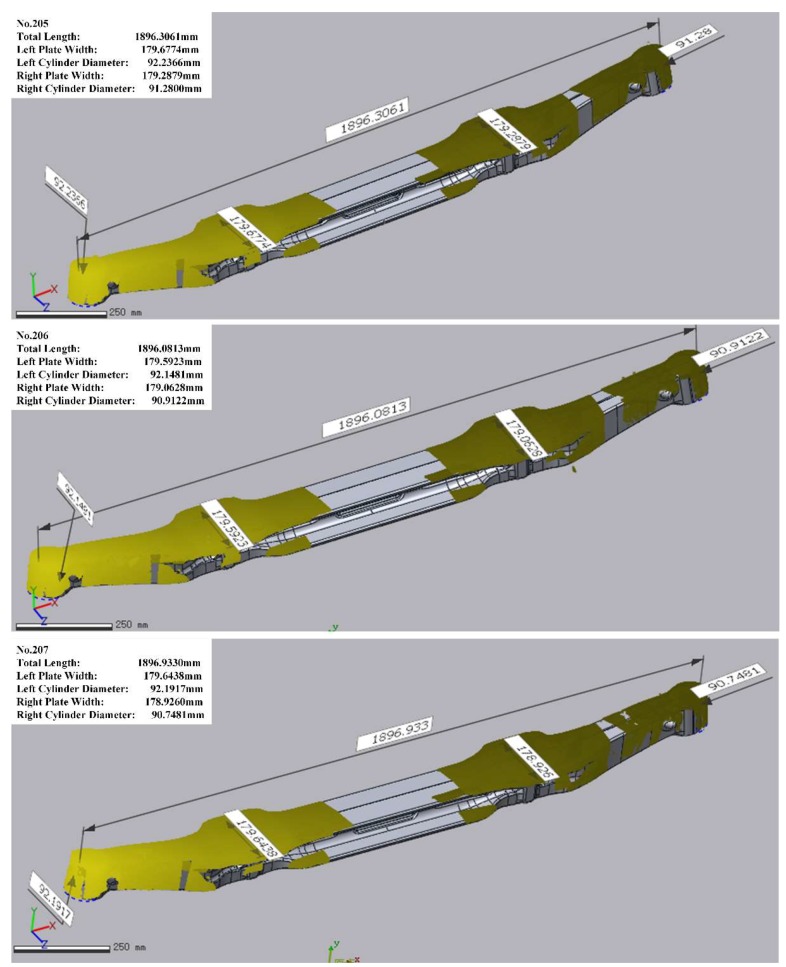
Inspection results of three thermal axles.

**Figure 15 sensors-18-04368-f015:**
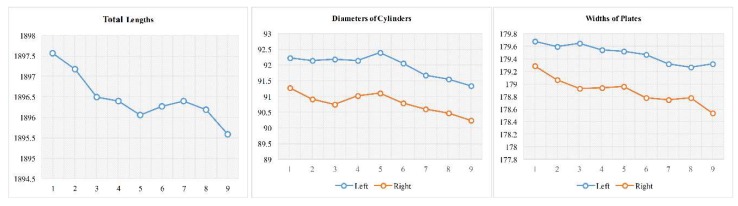
Trends of key dimensions.

**Table 1 sensors-18-04368-t001:** Results of 11 inspections of the same axle (unit: mm).

**Inspection Order**	**1**	**2**	**3**	**4**	**5**	**6**
Total length	1888.46	1888.35	1888.46	1888.37	1888.37	1888.46
Right plate width	177.45	177.44	177.43	177.43	177.42	177.43
Right cylinder diameter	88.48	88.48	88.48	88.45	88.46	88.45
Left plate width	178.87	178.87	178.87	178.87	178.87	178.87
Left cylinder diameter	89.71	89.70	89.71	89.70	89.71	89.70
**Inspection Order**	**7**	**8**	**9**	**10**	**11**	**Limit Deviation**
Total length	1888.46	1888.35	1888.46	1888.46	1888.37	0.11
Right plate width	177.45	177.44	177.45	177.43	177.43	0.03
Right cylinder diameter	88.47	88.48	88.46	88.48	88.45	0.03
Left plate width	178.87	178.87	178.87	178.88	178.88	0.01
Left cylinder diameter	89.71	89.70	89.70	89.71	89.70	0.01

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
