# Peer review of "A Robot-Driven 3D Shape Measurement System for Automatic Quality Inspection of Thermal Objects on a Forging Production Line"

_sensors, 2018, doi:10.3390/s18124368_

Round 1

Reviewer 1 Report

This paper proposes 3D structured light based system for quality inspection of thermal objects from automotive industry. 3D structured light head is mounted on robot arm to speed up and automate measurement process. This paper shows interesting application.

However I have some points that should be addressed by authors:

1)      Abstract: “matrix recording component” – it is unclear and not widely known term.

2)      Introduction: lines 75-87 – repetition from abstract. It will be more elegant if you just omit these sentences. You could write main points in bullet style.

3)      2.1.3 section:  Title is not good. In this chapter authors presents how to calculate transformation for a fixed set of measurement views. Maybe  title “View integration method” is better.

4)      2.1.3 section: I have large experience with using robot arm for 3D scanning. And I know that robot arm not always fit into exactly the same position. Please provide some validation that during different conditions (heat, etc.) measurement head is exactly in the same position – especially angular accuracy is important.

5)      2.1.3 section: Also regarding lines 209-217 you uses ICP after initial registration so simple calibration of 3D structured light and robot coordinates is enough from accuracy point of view. Why you use photogrammetric camera for such purpose?

6)      Is that mean that if you change measurement plan (number or position-orientation of 3D head) you have to repeat calibration with markers? Is it optimal solution vs robot arm – 3D head mutual calibration?

7)      202-206: Please be more clear. What means the allowable deviation? Is it deviation of whole reconstruction or just this algorithm (from section 2.2.1)? If whole how could you explain that 0,02mm is from this algorithm and what with the other influences like: error of alignment, error of measurement, etc.

8)      219-288: Your task for view registration is relatively simple. You could just before ICP filter out 3D data from background information and any good ICP based algorithm will work correctly. Am I right? Your ICP algorithm don’t shows any novelty in present form.

9)      3.2.1 Do you compare known distances of vehicle axle with your measurement results? If so please write it clearly and describe what is the source of reference data. If not please show such results. We don’t know how accurate system is? You can also in introduction specify how accurate measurement should be to work in practice.

10)   Section 4 should be Conclusions not Discussion. Please provide some limitations and critics of developed systems.

Reviewer 2 Report

This paper deals with a robot-driven 3D shape measurement system for 2 automatic quality inspection of thermal objects on a 3 forging production line. This is very interesting, and the proposed approach and methodology is presented in detail.

Some comments to the authors will be shown below:

The literature review is not adequate. The authors should enhance this section with papers mainly from the field of robot and robot control. Some indicative papers which fit the presented paper are the following:

a.       "Design and simulation of assembly systems with mobile robots", CIRP Annals-Manufacturing Technology, 2014.

b.       "An approach for implementing power and force limiting in sensorless industrial robots", 7th CIRP Conference on Assembly Technologies And Systems, CATS 2018.

c.       "A machine learning approach for visual recognition of complex parts in robotic manipulation", 27th International Conference on Flexible Automation and Intelligent Manufacturing, (FAIM2017) 27-30 June, Modena, Italy, Volume 11, pp. 423-430, 2017.

The authors presented the results of five inspections of the same axle as the results of the experiments. Firstly, the number of experiments are very small. You should provide more results in order to validate the methodology. Secondly, you did not mention the real dimension of the axle. You should compare the vision system results with the real dimensions.

The existing solutions are mentioned in the introduction of the paper. Although, the performance of the proposed approach should be compared with another during the experimental phase. Tables with results regarding the accuracy and speed of the proposed approach could be provided in comparison with the existing solutions.

The authors presented an online measurement system. Although, they did not mention anything about the time to execute the measurement online. How can the authors validate that this system works online?

The authors proposed a time-sharing technique for receiving data from the camera sensors. In the proposed method the cameras are not triggered together.  So, there is not an accurate relation between them. How this affects the technique that is proposed in the following chapters.

The ICP algorithm requires a proper initial value and the approximate registration of two-point clouds to prevent the algorithm from falling into local extremes. Can you descript how the initial estimation was generated?

Instead of the hand-eye calibration method, a marker-based technique was used. Why the marker-based technique is better than the original method? Which algorithm was used for the detection of the marker? How the data from the portable photogrammetry camera were aligned with the data from the real sensor? Can you descript the iteration process?

The implementation part is missing from the paper. Which software was used to implement the proposed methodology? Which is the data acquisition mechanism?

In Table 1, the units are missing.

Figures 2, 4, 5, and 10 have low resolution. Please fix them.

In equation 4, the font size of the numerators and denominators are quite small. Try to adjust them in order to be more visible to the reader.

The future plan is missing. What are the next steps to improve the proposed methodology? The authors should define the future activities.

Reviewer 3 Report

I recommend to review the criterion on line 272: "when ? is less than 0.256°", I think it should be "greater than".

Round 2

Reviewer 1 Report

I have no critical remarks and I appreciate the authors' work on the amendments. I recommend to publish in its current form.

Reviewer 2 Report

accept